# Influence of Sex and Body Size on the Validity of the Microsoft Kinect for Frontal Plane Knee Kinematics During Landings

**DOI:** 10.3390/s25175593

**Published:** 2025-09-08

**Authors:** Jillian Neufeld, Vital Nwaokoro, Derek N. Pamukoff

**Affiliations:** School of Kinesiology, Western University, 1151 Richmond St., London, ON N6A 3K7, Canada; jillian.neufeld@gmail.com (J.N.);

**Keywords:** knee, valgus, landing, sex, biomechanics

## Abstract

Three-dimensional (3D) motion capture is inaccessible, and the Microsoft Kinect is an alternative to measure surrogates of knee valgus that may contribute to anterior cruciate ligament (ACL) injury risk. We evaluated the influence of sex and body size on the agreement between methods. A total of 40 (10 per sex and BMI group) participants were included. The Kinect and motion capture measured knee ankle separation ratio (KASR) and knee abduction angles (KAAs). Intraclass correlation coefficients (ICCs) evaluated agreement between methods. 2 (sex) by 2 (BMI) by 2 (method) ANOVA compared kinematics between groups. Agreement between methods was moderate-to-good for KASR (initial contact ICCs 0.667–0.86; peak flexion ICCs 0.766–0.882). Agreement for KAA was low-to-moderate (initial contact ICCs 0.128–0.575; peak flexion ICCs 0.315–0.760). There was a BMI-by-method interaction for KASR at initial contact (*p* < 0.01) and a main effect of method (*p* < 0.01). There were BMI-by-method interactions for KAA (initial contact *p* > 0.01; peak knee flexion *p* < 0.01). The high BMI group had greater KAAs than the low BMI group, but only using motion capture. The Kinect is an alternative for measuring KASR, but not KAA. The high BMI group had greater KAAs than the low BMI group, but only when measured with motion capture.

## 1. Introduction 

Anterior cruciate ligament (ACL) injuries have a high risk of re-injury [1]. Dynamic knee valgus during landing is a multi-joint and multi-planar mechanism of ACL injury that involves hip internal rotation and adduction, knee abduction, tibial external rotation, and foot pronation and external rotation [2]. As such, the Drop Vertical Jump (DVJ) is commonly used to assess biomechanics during landing [2,3]. Furthermore, frontal plane knee kinematics during a DVJ have been associated with ACL injury [2,4]. Movement patterns during landing have been modified through the implementation of neuromuscular training programs [5]. Therefore, evaluating frontal plane knee kinematics during a DVJ may identify individuals who would benefit from preventative neuromuscular exercise programs for decreasing ACL injury risk [6,7]. The gold standard for measuring knee kinematics is 3-dimensional (3D) motion capture [8]. However, 3D motion capture is inaccessible for clinicians due to cost and training requirements [6,8]. Alternatives to 3D motion capture include monocular event cameras and convolutional neural networks (CNNs) [9]. Monocular event cameras may require active markers such as flashing LEDs to identify body landmarks, thus having similar limitations to marker-based 3D motion capture [10]. Video analyses with CNNs have shown promising results for human pose estimation and motion analysis. However, CNNs have high computational and memory costs and require large training data sets for optimal accuracy and performance [9,11].

The Microsoft Kinect (V2.0, Microsoft, Redmond, WA, USA) was developed for the Xbox 360 game console, and ACL Gold software was developed to measure knee movement during DVJs with good reliability and validity [6]. The Kinect with ACL Gold measures the knee ankle separation ratio (KASR) and knee abduction angle (KAA) at initial contact (IC) of the feet on the floor and at the point of peak knee flexion (PKF) during a DVJ [6]. The KASR is the ratio of the distance between the knees relative to the distance between the ankles and provides a surrogate measurement for knee abduction [12]. The KAA is the relative angle between the thigh and shank in the frontal plane [13]. Medial knee positioning identified based on the KASR and KAA is associated with dynamic knee valgus and ACL injury [2,4]. Therefore, KASR and KAA measured with the Microsoft Kinect may be useful for analyzing landing biomechanics in clinical settings.

Analyzing landing biomechanics is important for individuals at greater risk of ACL injury, such as individuals with high BMI or females, while participating in physical activity [2,14]. Females are 4–6 times more likely to sustain an ACL injury than males [2], which may be due to modifiable characteristics like movement patterns [15]. Females land with less knee flexion and more knee abduction than males, which may place additional stress on the ACL [2]. Secondly, greater body mass is associated with greater peak tibiofemoral compressive force [16,17], which places stress on the ACL during landing [14]. Greater body size is also correlated with increased body surface area [18], which may influence motion capture analyses [19]. The Microsoft Kinect uses a scan of body surface area to identify limbs and joint centers [20]. Differences in body surface area and body mass distribution may influence accuracy in joint center identifications [19,20]. Finally, there may be an interaction between sex and body size on landing biomechanics due to body mass distribution and a stronger correlation between body size and body fat percentage in females than in males [21]. Females in all body mass index (BMI) categories have more soft tissue around the hips compared to males of equivalent BMI classification [22], which may influence joint center identification [20]. Therefore, sex and body size could influence the reliability and validity of the Microsoft Kinect [19,20], which were not considered in previous evaluations of validity [6].

The purpose of this study was to evaluate the influence of sex and body size on the validity of the Microsoft Kinect for measuring frontal plane knee motion during landing. The primary hypothesis was that knee valgus characteristics during the DVJ assessed using the Microsoft Kinect would be similar to motion capture for all sex and body size groups. A secondary purpose was to compare knee valgus characteristics during landing between males and females and those with high and low BMI using both methods. We hypothesized that females and those with high BMI would demonstrate greater knee valgus during landing than males and those with low BMI, respectively.

## 2. Materials and Methods

### 2.1. Study Design

This was a controlled cohort study. A single testing session involved anthropometric measurements, a warm-up, and a Drop Vertical Jump (DVJ) protocol while knee kinematics were simultaneously recorded by 3D motion capture and the Microsoft Kinect.

### 2.2. Participants

Fifty-five individuals were screened to recruit the forty participants (10 low BMI females, 10 high BMI females, 10 low BMI males, and 10 high BMI males) who participated in this study (Table 1). Sample size was calculated based on an expected intraclass correlation (ICC) of 0.84, 5 measurements per participant, and a minimum acceptable ICC of 0.5 (β = 0.2, α = 0.05). The expected ICC was based on previous research evaluating the validity of the Kinect relative to motion capture [6]. Additionally, 9 participants per group were required to achieve 80% power based on large effect size differences between males and females [15] and between those with high and low BMI [23] in a mixed factor ANOVA design (f = 0.3, β = 0.2, α = 0.05; G.Power 3.1.9.7). Participants were 18 years of age or older, able to read and comprehend English, at a minimum recreationally active, and in a BMI category between 18.5 kg/m^2^ and 24.9 kg/m^2^ or >30.0 kg/m^2^. Physical activity was self-reported and at least three times weekly. Exclusion criteria included lower body injury within 6 months of participation, history of lower body surgery, and ongoing knee pain. Participants provided written informed consent prior to participation, and all methods were approved by the University Research Ethics Review Board (Western University HSREB 120801).

Participants’ height (m) and mass (kg) were measured using a digital scale and wall-mounted stadiometer, respectively, to determine body mass index [$BMI=\frac{mass\;(kg)}{{height}^{2}(m)}$]. Subjects were allocated to either the low BMI group [18.5–24.9 kg/m^2^] or the high BMI [>30 kg/m^2^]. Individuals whose BMI fell between 25.0 and 29.9 kg/m^2^ were excluded to ensure a difference in body size between groups.

### 2.3. Procedures

An 8-camera motion capture system recording at 200 Hz (Qualisys, Gothenburg, Sweden) and force plates recording at 2000 Hz (AMTI, Watertown, MA, USA) were used for the 3D motion capture measurements. Following a general warm-up of body weight squats, lunges, and jumping jacks, participants were outfitted with reflective markers. Calibration markers were placed bilaterally on the iliac crests, greater trochanters, medial and lateral epicondyles of the knees, medial and lateral malleoli, the base of the first and fifth metatarsals, and the calcanei. Rigid marker clusters of 4 non-colinear reflective markers were affixed to the sacrum and bilaterally on the thighs, shanks, and dorsum of the feet for dynamic tracking [8]. The use of rigid clusters increases the reliability of 3D motion capture in high BMI groups [24]. The Microsoft Kinect was positioned on a tripod 2.5 m in front of the force plates, perpendicular to the anterior–posterior axis to allow measurement within the frontal plane [6].

Following marker placement, participants underwent a standing calibration trial and removal of the calibration markers [8]. The researcher verbally explained the DVJ task using standardized instructions and provided a visual demonstration. The DVJ task involved dropping off a 30 cm box positioned 10 cm from the force plates, landing with both feet, immediately performing a maximal vertical jump, landing with both feet simultaneously, and then taking 3 steps forward [6]. Following instructions and demonstration, the participant completed 3 practice DVJs to familiarize themselves. The Microsoft Kinect and 3D motion capture systems simultaneously recorded 5 DVJ trials per participant, but landing events (IC and PKF) were independently determined per method rather than synchronized. All data from trials not successfully captured using both methods were discarded and reattempted. A successful trial for the Kinect was where the ACL Gold software identified all points of interest and there was no visibly evident errors such as airborne feet. A successful 3D motion capture trial tracked all markers throughout the task, and both feet made full contact with separate force plates. 3D motion capture data and still frames from the Kinect of IC and PKF events were visually examined for missing marker data or event identification errors. No marker tracking errors were identified, and Kinect trials with errors in event identification or pose estimation were discarded.

### 2.4. Data Reduction

Microsoft Kinect data were analyzed using ACL Gold software [6], which measures KASR as follows:$$KASR=\frac{Rknee\;position-Lknee\;position}{Rankle\;position-Lankle\;position}$$

A KASR of 1 indicates equal distance between the knees and ankles, which suggests that the knees are directly superior to the ankles, less than 1 indicates a knee position medial to the ankles, and greater than 1 indicates a knee position lateral to the ankles [6]. The Kinect also measures KAA, defined as the angle of the shank with reference to the thigh in the frontal plane [13], at IC and PKF during landing [6]. The Kinect identifies IC as the first frame where the ankle joint center velocities decrease and PKF as the frame where the hip joint centers and base of the spine locations are lowest with reference to the floor [25]. The thigh segment is defined as the length between the hip and knee joints, the shank as the length between the knee and ankle [25]. Joint centers are identified as a function of information from the depth sensor and two-dimensional camera [26]. The depth sensor uses infrared beams and a time-of-flight sensor to gauge the distance between the Kinect and objects within the camera’s field [26,27]. The information from the depth sensor and camera is input into a decision tree matrix that estimates which pixels correspond with joint center locations [26,28]. Biomechanical analyses typically use the average of trials to quantify habitual movement patterns [29]. Therefore, average KASR scores and KAA were calculated for each participant from the 5 successful trials. Similarly, the average KAA was calculated for each leg separately. As no difference was found between right and left knee abduction angles, right leg values were used for analysis.

Model construction and data analysis from 3D motion capture were completed using Visual 3D (v2021, HAS-Motion, Kingston, ON, Canada). Marker trajectories and force plate data were low-pass filtered at 12 Hz. KASR and KAA values were extracted at IC and PKF during each landing. IC was determined when force plates exceeded 20 N. The PKF was determined by the sagittal plane knee angle between the thigh and shank. The knee abduction angle was defined using Euler/Cardan angles (XYZ rotation sequence) as motion of the tibia relative to the femur in the frontal plane. The hip joint center was estimated as one-quarter the distance between the greater trochanter markers. The knee and ankle joint centers were defined as the midway point between the lateral and medial femoral epicondyle markers and the lateral and medial malleoli markers, respectively. Knee and ankle joint centers were used to calculate KASR according to the same equation as the Kinect.

### 2.5. Statistical Analyses

Mean and standard deviation were calculated for all demographics and for KASR and KAA by group and method. Data were inspected for normality using the Shapiro-Wilk test, and outliers were evaluated using boxplots. Agreement between methods was estimated using the intraclass correlation coefficient (ICC2, k) with 95% confidence intervals within each group (Table 2). ICC values were interpreted as <0.5 as poor, 0.50–0.75 as moderate, 0.76–0.90 as good, and 0.90–1.00 as excellent [30].

The influence of method, sex, and BMI was evaluated using a 2 (sex) by 2 (BMI) by 2 (method) ANOVA (α = 0.05) for KASR and KAA at IC and PKF. Post hoc comparisons evaluated significant interactions using a Bonferroni adjustment. Bland–Altman plots were generated to visualize systematic bias (Figure 1 and Figure 2). Scatterplots and linear association (R^2^) were derived to further visualize agreement between methods for each group (Appendix A). We compared peak knee flexion angles measured with motion capture between groups to interpret findings related to the hypotheses (Appendix A).

## 3. Results

Males and females with high body size had greater mass and BMI than males and females with low body size, respectively, and males were taller than females (Table 1). One outlier was identified in the high BMI male group in KAA at PKF during 3D motion capture analyses. After inspection of the biomechanical model, it was concluded that there were no errors and the values were biologically plausible based on published data [12]. Analyses were conducted with and without the outlier, and the interpretation did not differ. Therefore, we retained the individual as per the intended analyses.

Agreement between the methods for KASR was moderate-to-good at IC (ICC Range: 0.667–0.861, Table 2) and PKF (ICC Range: 0.766–0.882, Table 2). Scatterplots indicated weak-to-strong linear associations for KASR measurements at IC (R^2^ Range: 0.044–0.593, Appendix A) and at PKF (R^2^ Range: 0.107–0.874, Appendix A). KAA showed low-to-moderate agreement between methods for all groups at IC (ICC Range: 0.128–0.575, Table 2) and PKF (ICC Range: 0.315–0.760, Table 2). Scatterplots also indicated weak-to-moderate linear associations for KAA measurements at IC (R^2^ Range: 0.011–0.163, Appendix A) and at PKF (R^2^ Range: 0.088–0.412, Appendix A). Bland–Altman plots illustrate the average differences between methods per sex and body size group for KASR and KAA at IC and PKF (Figure 1 and Figure 2). Average differences between methods greater than zero indicate that the Kinect overestimated KASR at PKF.

Ensemble average waveforms for knee abduction angles during the DVJ are presented per group for visualization purposes (Figure 3), and data are presented in Table 3.

There was a significant BMI-by-method interaction (F_1,36_ = 10.022, *p* < 0.01, partial η2 = 0.218, Appendix A) for KASR at IC. Post hoc analyses indicated a greater difference between methods in the high BMI group (mean difference = 0.234 (0.191, 0.227) d = 2.52) than in the low BMI group (mean difference = 0.139 (0.096, 0.182), d = 1.09). There was a significant main effect of method on KASR at PKF (F_1,36_ = 15.17, *p* < 0.01, partial η2 = 0.297), indicating a lower KASR when measured using 3D motion capture compared to the Kinect (mean difference = −0.090 (−0.137, −0.043), d = 1.23).

There was a significant method by BMI interaction effect (F_1,36_ = 7.171, *p* = 0.01; partial η2 = 0.167). When collapsed across sex, post hoc analyses indicated that the high BMI group had a larger KAA at IC compared with the low BMI group (mean difference = 6.272 (3.778, 8.765), *p* < 0.01, d = 1.61), but only when evaluated using motion capture. Moreover, KAA was greater when measured using motion capture compared with the Kinect (mean difference = 5.083 (2.177, 7.989), *p* < 0.01, d = 1.11), but only in the high BMI group.

There was a BMI group-by-method interaction for the KAA at PKF (F_1,36_ = 12.985, *p* = 0.05, partial η2 = 0.10). When collapsed across sex, post hoc analyses indicated larger KAA at PKF in the high compared with the low BMI group (mean difference = 7.492 (2.733, 12.251), *p* < 0.01, d = 1.00), but only when measured using motion capture.

When collapsed across sex, the high BMI group had lower peak knee flexion angles than the low BMI group (−72.666 (9.533) vs. −87.633 (13.225), *p* < 0.01, d = 1.13; Appendix A).

## 4. Discussion

Dynamic knee valgus during landing is a multi-joint and multi-planar movement pattern associated with risk of ACL injury [2], and surrogate measurements include the KASR and KAA [6]. This study measured KASR and KAA across sex and BMI groups to assess agreement between methods and found moderate-to-good agreement between the Kinect and motion capture for KASR but low agreement for KAA, which partially supports our hypotheses. Furthermore, this study found effects of body size on measurement methods, which also supports our hypothesis. The secondary purpose was to assess the influence of sex and BMI on landing patterns. The high BMI group showed a more medial knee position during landing than the low BMI group when collapsed across sex and measured using 3D motion capture, supporting the hypothesis of differences in landing patterns between BMI groups. No differences were found in the landing patterns between males and females, which did not support our hypothesis.

The level of agreement between the Kinect and 3D motion capture was moderate-to-good for KASR for all groups. Knee separation distance has been shown to be predictive of average bilateral knee abduction angles [7] and may be a useful surrogate for measuring medial knee displacement to identify individuals at increased risk of ACL injury [2]. The moderate-to-good agreement we found between the Kinect and 3D motion capture is in line with previous validation studies [6,25] and supports the use of the Kinect in clinical settings for measuring KASR. KASR has been associated with movement patterns predictive of ACL injury [7], but clinically important differences in KASR have not been established.

Despite moderate-to-good agreement between methods, the difference in mean measurements between methods indicates that the Kinect may overestimate KASR (i.e., less medial knee displacement). A potential source of error is the participants’ limb or body position relative to the camera from a frontal plane perspective. Subtle transverse plane rotation or deviation from a perpendicular view may confound the estimation of joint center locations in the frontal plane, which is not a limitation of 3D motion capture. The consistent overestimation of the KASR should be considered when interpreting the Kinect measurements to avoid missing individuals with higher-risk landing patterns.

The overestimation of KASR may be greater in the high BMI group due to inaccuracies in joint center estimations [20] due to body surface area and mass distribution [18]. Body mass index influenced the level of agreement between the methods. We found better agreement for the low compared with the high BMI group. The Kinect uses a scan of body surface area to identify joint centers [20]. Previous research comparing joint center identification found a > 60 mm difference between methods for all lower body joints [20]. Greater body surface area associated with higher BMI [31] may contribute to error in joint center identification and lower agreement between methods in high compared with low BMI groups [20]. In an analysis of body surface area compared to BMI, Mance et al. found that as BMI increases, the limbs account for a greater percentage of body surface area compared to the trunk [31]. The difference in body surface area distribution between low and high BMI individuals may influence the accuracy of the Kinect [31]. Interpretation of Kinect measurements should consider the influence of BMI, and 3D motion capture may be necessary for participants with larger body sizes to avoid errors in injury risk stratification. Interestingly, the high BMI group also required more attempts to collect 5 acceptable trials. Repeated trials were due to an inability of the Kinect to identify IC or PKF, which suggests difficulty tracking the joint centers needed for identifying events. Repeated attempts may contribute to fatigue or a learning effect and should be considered when implementing the Kinect in practice.

Interpretation of Kinect-derived measurements should also take into consideration the wide confidence intervals for ICC point estimates between the Kinect and 3D motion capture. Wide confidence intervals, including negative values, suggest uncertainty in the true agreement between methods. Practitioners should consider the potential for higher or lower agreement between the Kinect and gold-standard measurement when implementing testing with the Kinect. Furthermore, readers should consider the variability in agreement between the Kinect and 3D motion capture when interpreting other research reporting Kinect-derived measurements. In our study, the groups with the narrowest confidence intervals were low BMI females at IC and low BMI males at PKF when measuring KASR, suggesting that the true agreement between methods may be in the moderate-to-good range for these groups and timepoints.

Previous research temporally aligned the systems for event identification, which may contribute to higher agreement between methods compared with our data [25]. However, event identification using force plates and 3D motion capture would not be feasible in a clinical setting, and it is important for determining the validity of the Kinect. Therefore, we chose not to synchronize events between methods to better reflect measurement precision in clinical environments. Practitioners should be aware of potential errors introduced by discrepancies in event identification when interpreting Kinect-derived measurements. Future studies should consider analyses on both synchronized and non-synchronized data to better understand sources of error.

There was low-to-moderate agreement between methods for KAA, and weak linear associations demonstrate that the Kinect does not consistently over- or underestimate KAA. Kinect sensor perspective and the participant’s joint positioning at different points during landing may contribute to low agreement [20]. Hip rotation and ankle pronation contribute to the appearance of knee abduction from a frontal plane perspective [13], like that of the Kinect. Thus, an inability to measure KAA due to joint motion in other planes may contribute to low agreement between methods. Furthermore, any rotation out of the frontal plane during the DVJ may influence measurements due to the perspective of the sensor [20,26]. In contrast, 3D motion capture identifies joint positions in all three planes and is not reliant on a single camera [8]. Thus, KAA measured with 3D motion capture is not influenced by hip or ankle positioning nor by rotation during the landing relative to a single camera [8].

The Kinect’s perspective may further contribute to inaccuracy in measuring KAA if hip joint centers become obstructed from the sensor’s frontal plane view. The landing process utilizes hip and knee flexion to attenuate ground reaction forces [2,32]. The hip joints may become obstructed from the Kinect by the knees or shanks during the deepest portion of the landing. Previous comparison of joint center identification between methods showed >100 mm of difference in hip joint center locations during deep knee flexion [20]. As such, the Kinect may not accurately identify the thigh segments to measure KAA at PKF [20].

Conversely, good agreement between the methods for the high BMI male group at PKF may reflect a different landing strategy between BMI groups. Stiff landings are characterized by lesser knee flexion [2] than soft landings, and the hip joints may not descend behind the knees or shanks. In our study, the group with a high BMI had less knee flexion compared with the low BMI group. Thus, thigh segment identification and KAA measurements may be more accurate, whereby the hip joints stay visible to the Kinect sensor throughout the landing. Despite acceptable agreement between methods for KAA in high BMI males, low agreement for KAA for other groups suggests that Kinect-derived KAA may lead to misidentification of athletes at risk of ACL injury. This is supported by previously reported inaccuracies in joint center identification at PKF [20]. We did not identify a difference in knee valgus characteristics during a DVJ between females and males in our analysis. However, previous research has identified differences between males and females in landing patterns and ACL injury incidence [2,15]. Differences between male and female landing patterns may be more evident in single limb landings [4] or in other kinematic variables such as hip rotation [33], which were not evaluated in this study. Future projects may consider developing and validating software for the Kinect for other movement patterns, such as single limb landings.

## 5. Limitations

There are limitations to consider when interpreting the results of this study and when using the Kinect and ACL Gold software. Firstly, there is an expected error due to soft tissue artifact in 3D motion capture [19]. Soft tissue artifact is higher in participants with a high BMI, particularly around the pelvis and thigh [19,31]. We utilized rigid marker clusters affixed to the thighs, shanks, and sacrum to reduce motion of markers relative to each other [19], but motion artifact likely contributed to greater error in participants with a high BMI [31]. We also excluded participants with 25.0–29.9 kg/m^2^ BMI to ensure a difference in body size between groups, as there can be considerable variation in body composition between individuals of similar BMI [21]; however, this may influence the generalizability of our findings. Moreover, the high BMI group required more attempts during data collection. Repeated attempts increase the time demand on the participant and tester and increase physical strain on the participant. Body size and soft tissue are relevant to both methods by contributing to error in marker placement and soft tissue artifact [19] and influencing body surface area [20,31]. Thus, body size is an important consideration when interpreting data from both methods. Secondly, joint center identification may be influenced by environmental conditions such as lighting or background motion. Our study was conducted in a well-lit, controlled laboratory environment, and testing with the Kinect should be conducted with a static background and adequate lighting to minimize interference.

We also note that landing scenarios during sport may include unilateral or staggered landings and be accompanied by cognitive distractions not accounted for in our DVJ testing [4]. Further software development to expand the measurement capabilities of the Kinect may broaden its applicability to ACL injury risk screening.

Our study screened participants for minimum physical activity participation, but not maximum, and diverse sport participation may account for findings that differ from other studies that use more homogenous groups, such as high school [1] or college athletes [2]. Future research may consider controlling maximum activity levels or stratifying participants by athletic ability to control for potential differences in landing patterns and determine the validity of the Kinect across athletic ability levels.

## 6. Conclusions

The Microsoft Kinect with ACL Gold software had moderate-to-good agreement with 3D motion capture for measuring KASR at IC and PKF for all sex and body size groups. The Kinect had the best agreement with 3D motion capture with ICCs in the good range for all females at IC, all males at PKF, and high BMI females at PKF. This supports the use of the Kinect as a measurement tool for measuring KASR during landing, although wide confidence intervals suggest some uncertainty in agreement between methods across participants. Conversely, the Kinect had low-to-moderate agreement for measuring KAA. Body size, but not sex, influenced agreement between the Kinect and 3D motion capture, particularly for measuring KAA. The high BMI group, regardless of sex, had greater KAA during landing than the low BMI groups, but only when measured with 3D motion capture. The Kinect underestimated the KAA compared to 3D motion capture for females and males with high BMI at IC and PKF. The underestimation of KAA in the high BMI groups caused the Kinect to not identify the different landing patterns between high and low BMI groups that were identified with motion capture. Overestimated KASR or underestimated KAA both suggest that the participant may be at a lower risk of ACL injury than they truly are. Future research to establish benchmarks for injury risk classifications should consider Kinect’s tendency to minimize injury risk.

## Figures and Tables

**Figure 1 sensors-25-05593-f001:**
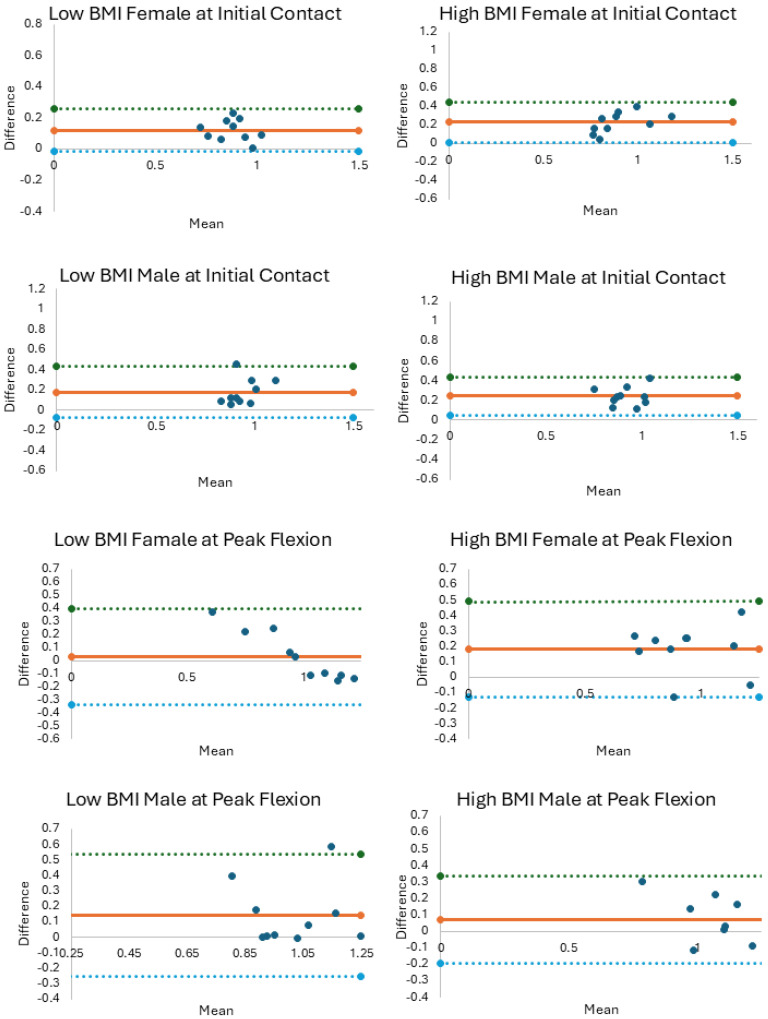
Bland–Altman plots for knee–ankle separation ratio plotting the average of the 3D motion capture and Kinect measurements (x axes) by the difference between measurement methods (y axes). The solid orange line represents the average difference between measurement methods with the upper 95% CI (dotted green line) and lower 95% CI (dotted blue line).

**Figure 2 sensors-25-05593-f002:**
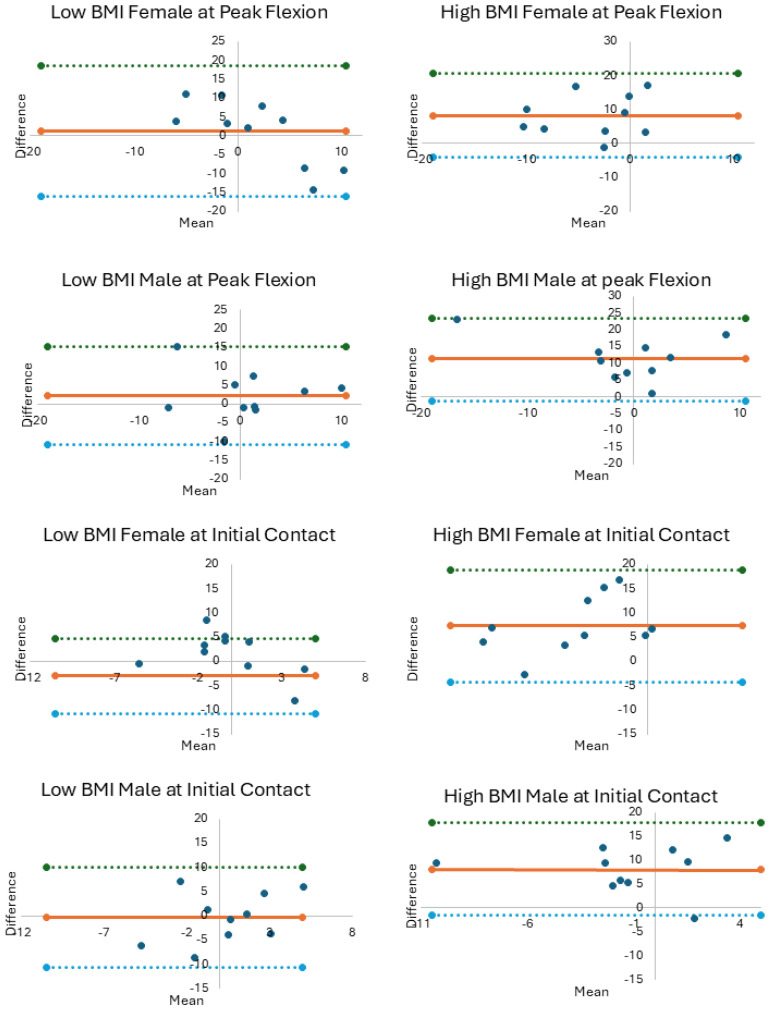
Bland–Altman plots for knee abduction angle plotting the average of the 3D motion capture and Kinect measurements in degrees (x axes) by the difference between measurement methods (y axes). The solid orange line represents the average difference between measurement methods with the upper 95% CI (dotted green line) and lower 95% CI (dotted blue line).

**Figure 3 sensors-25-05593-f003:**
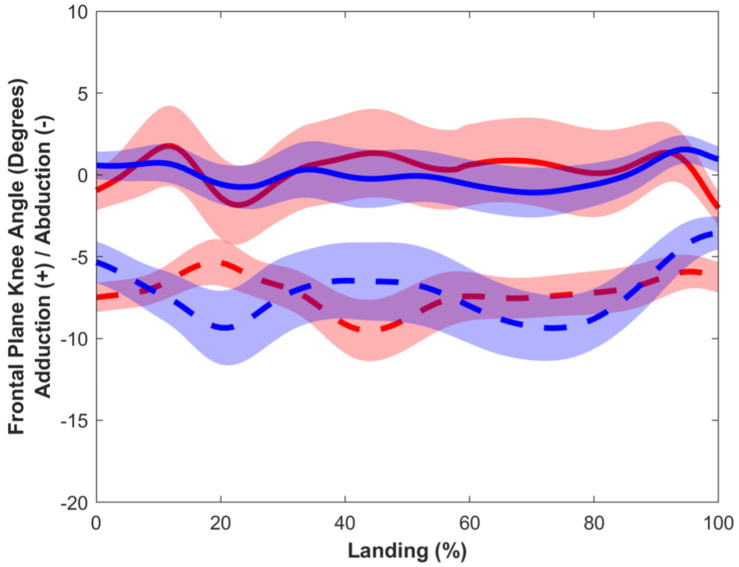
Ensemble average waveform of knee abduction angle from motion capture (red indicates female; blue indicates male; solid lines indicate low BMI; dashed lines indicate high BMI). Shaded regions indicate 95% confidence interval.

**Table 1 sensors-25-05593-t001:** Demographic information (mean ± SD).

	Low BMI Females (n = 10)	High BMI Females (n = 10)	Low BMI Males (n = 10)	High BMI Males (n = 10)
Age (years)	20.9 ± 2.6	20.8 ± 2.0	21.6 ± 1.3	21.0 ± 1.5
Height (m)	1.70 ± 0.07	1.67 ± 0.10	1.81 ± 0.07	1.85 ± 0.07
Mass (kg)	65.24 ± 4.85	97.89 ± 17.75	77.31 ± 7.42	118.30 ± 15.78
BMI (kg/m^2^)	22.52 ± 1.07	34.99 ± 4.40	23.51 ± 1.38	34.56 ± 3.18

Note: Low BMI is 18.5–24.9 kg/m^2^**;** high BMI indicates BMI > 30 kg/m^2^.

**Table 2 sensors-25-05593-t002:** Agreement between methods (ICC2,k [95% Confidence Interval]).

	Low BMI Females	Low BMI Males	High BMI Females	High BMI Males
KASR at IC	0.861 [0.442, 0.966]	0.667 [−0.340, 0.917]	0.831 [0.319, 0.958]	0.728 [−0.095, 0.932]
KASR at PKF	0.766 [0.059, 0.942]	0.882 [0.525, 0.971]	0.806 [0.218, 0.952]	0.805 [0.217, 0.952]
Knee abduction angle at IC	0.360 [−1.576, 0.841]	0.172 [−2.333, 0.794]	0.128 [−2.509, 0.783]	0.575 [−0.713, 0.263]
Knee abduction angle at PKF	0.315 [−1.759, 0.830]	0.582 [−0.681, 0.896]	0.533 [−0.879, 0.884]	0.760 [0.033, 0.940]

Note: IC = initial contact; PKF = peak knee flexion.

**Table 3 sensors-25-05593-t003:** Mean (SD) of knee abduction and knee–ankle separation ratio by group.

	Low BMI Females (n = 10)		Low BMI Males (n = 10)		High BMI Females (n = 10)		High BMI Males (n = 10)	
	Kinect	Motion Capture	Kinect	Motion Capture	Kinect	Motion Capture	Kinect	Motion Capture
KASR IC (0–2.0)	0.938 (0.092)	0.817 (0.106)	1.03 (0.121)	0.872 (0.055)	1.012 (0.175)	0.786 (0.117)	1.04 (0.109)	0.799 (0.101)
KASR PKF (0–2.0)	0.995 (0.109)	0.964 (0.286)	1.044 (0.120)	0.965 (0.136)	1.032 (0.194)	0.851 (0.199)	1.130 (0.132)	1.060 (0.197)
Knee Abduction IC (°)	1.332 (3.270)	−0.907 (4.451)	0.244 (4.634)	0.610 (3.022)	−0.170 (5.210)	−7.53 (3.250)	2.842 (4.462)	−5.31 (4.576)
Knee Abduction PKF (°)	2.416 (3.283)	1.251 (9.241)	1.630 (5.972)	−5.167 (6.259)	0.524 (5.980)	−7.695 (5.125)	4.781 (6.130)	−6.555 (8.333)
DVJ attempts (n)	7.8 (2.616)		8.8 (4.264)		7.8 (2.573)		13 (3.590)	

Note: IC = initial contact; PKF = peak knee flexion. Knee abduction angles were measured in degrees (− indicates more abducted). DVJ attempts = average number of attempts required to collect 5 successful trials.

## Data Availability

Supporting data can be provided upon reasonable request.

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
