# Peer review of "Influence of Sex and Body Size on the Validity of the Microsoft Kinect for Frontal Plane Knee Kinematics During Landings"

_sensors, 2025, doi:10.3390/s25175593_

Round 1

Reviewer 1 Report

Comments and Suggestions for Authors

This manuscript systematically evaluates the accuracy of motion analysis using Microsoft Kinect measurements in Drop Vertical Jump tasks. The comparison with ground truth measurements obtained from an 8-camera motion capture system across four subject groups classified by sex and BMI revealed that the Kinect measurements demonstrated moderate-to-good accuracy in KASR (knee–ankle separation ratio) at IC and PKF, but low-to-moderate agreement in KAA (knee abduction angles). Although the results may serve as useful evidence for the clinical application of the Kinect, the current manuscript has several issues that should be addressed before publication. Please consider the following points:

- Recent advancements in 2D and 3D motion analysis using monocular cameras and CNN-based image recognition techniques have demonstrated good accuracy compared with the Kinect. However, the authors do not mention these novel techniques in the introduction or discussion. Please clarify the advantages and disadvantages of the Kinect relative to these methods, and add appropriate references.

- In Figures 1 and 2, the x- and y-axes should be aligned to the left and bottom, respectively, as the axis labels are difficult to read due to overlapping with the plots. Please also add an explanation of the three horizontal lines in these figures to the legends.

- The resolution of Figures 1, 2, S1, and S2 should be improved.

- Please consider adding the Kinect measurement results to Figure 3 to facilitate a better understanding of the nature of the estimation error.

Author Response

Comments from Reviewer 1

- Recent advancements in 2D and 3D motion analysis using monocular cameras and CNN-based image recognition techniques have demonstrated good accuracy compared with the Kinect. However, the authors do not mention these novel techniques in the introduction or discussion. Please clarify the advantages and disadvantages of the Kinect relative to these methods and add appropriate references.

Response: Thank you for identifying the gaps in the background involving other methods of motion capture. We have added to the background section to include discussion of monocular and CNN tools for motion analysis in lines 38-44 to read “Alternatives to 3D motion capture include monocular event cameras and convolutional neural networks (CNNs). Monocular event cameras may require active markers such as flashing LEDs to identify body landmarks, thus have similar limitations to marker-based 3D motion capture. Video analyses with CNN have shown promising results for human pose estimation and motion analyses. However, CNNs have high computational and memory costs and require large training data sets for optimal accuracy and performance.”

- In Figures 1 and 2, the x- and y-axes should be aligned to the left and bottom, respectively, as the axis labels are difficult to read due to overlapping with the plots. Please also add an explanation of the three horizontal lines in these figures to the legends.

- The resolution of Figures 1, 2, S1, and S2 should be improved.

Response: Thank you for identifying the ways we could improve figures 1 and 2. We have moved the axes to improve readability, added axis descriptors and identified the solid and dotted lines in each chart in the figure legends (see below). We note that resolution was downgraded in the submission process, but confirm that uploaded versions for typesetting are at 300dpi.

The caption for figure 1 (lines 189-192) has been expanded to “Bland Altman Plots for Knee-Ankle Separation Ratio plotting the average of the 3D motion capture and Kinect measurements (x axes) by the difference between measurement methods (y axes). The solid orange line represents the average difference between measurement methods with upper 95% CI (dotted green line) and lower 95% CI (dotted blue line).” Similarly, the caption for figure 2 (lines 195-198) has been expanded to “Bland-Altman Plots for Knee Abduction Angle plotting the average of the 3D motion capture and Kinect measurements in degrees (x axes) by the difference between measurement methods (y axes). The solid orange line represents the average difference between measurement methods with upper 95% CI (dotted green line) and lower 95% CI (dotted blue line).”

- Please consider adding the Kinect measurement results to Figure 3 to facilitate a better understanding of the nature of the estimation error.

Response: We agree that being able to visualize both methods on the Figure 3 waveform would be beneficial. However, the Kinect with ACL Gold only provides discrete measurements for KAB at initial contact and peak knee flexion. No time series data is provided to generate a similar waveform.

Reviewer 2 Report

Comments and Suggestions for Authors

Review comment  
Thank you for the opportunity to review this manuscript submitted to Sensors.  
The study is well-structured and addresses a relevant topic in clinical and sports biomechanics. However, there are several methodological and interpretative issues that need to be addressed before the manuscript can be considered for publication.  

Specific comments  
1. M2.3, lines 118–120: How do the authors account for the potential measurement bias introduced by identifying IC and PKF independently in the two systems without synchronization?  
2. M2.2, lines 94–97: Why were participants in the mid-range BMI category (25.0–29.9 kg/m²) excluded, and how does this affect the generalizability of the findings?  
3. R-Table2, lines 166–167; R-text, lines 188–197: For KAA measurements with ICC values below 0.5, was any independent verification (e.g., video-based assessment) performed to confirm accuracy?  
4. M2.2, lines 83–85: Given that the analysis involved a 2×2×2 ANOVA, how was the sample size calculation adjusted to ensure adequate power for detecting interaction effects?  
5. R-Table2, lines 166–167: How do the authors interpret ICC values with such wide confidence intervals, some spanning from negative values to high positive values?  
6. C, lines 343–354: Can the authors clarify under which specific conditions the Kinect tends to underestimate injury risk, rather than presenting this as a universal outcome?  
7. M2.2, lines 86–90; D, lines 308–312: How applicable are these findings to elite athletes or high ACL-risk populations, given that the sample consisted of recreationally active adults?  

Author Response

Comments from Reviewer 2

  1. M2.3, lines 118–120: How do the authors account for the potential measurement bias introduced by identifying IC and PKF independently in the two systems without synchronization?  

Response: We agree that not synchronizing the identification of IC and PKF introduces potential for measurement bias into our measurements. Accurate identification of IC and PKF is necessary for obtaining accurate KAA and KASR measurements and since the Kinect would not utilize force plates for event identification in clinical practice. Therefore, we elected to compare the outputs of each system fully independently to obtain a representative magnitude of difference between the motion capture methods as they are typically used (i.e. not concurrently). Lines 303-311 discuss our choice to not synchronize event identification, and we have added a sentence (lines 308-309) to encourage practitioners to consider event identification as a potential source of error when interpreting data. 

  1. M2.2, lines 94–97: Why were participants in the mid-range BMI category (25.0–29.9 kg/m²) excluded, and how does this affect the generalizability of the findings?  

Response: Thank you for identifying the potential implications of excluding the middle range of BMI (25.0-29.9 kg/m2). We elected to exclude this group to ensure sufficient difference between groups. There are known differences in landing biomechanics between those with high and low BMI with large effect sizes. Your fourth comment regarding power for the ANOVA analysis is related to this recruitment choice. Excluding the mid-range of BMI provided confidence that we could reasonably expect large effect size differences between body size groups. Nonetheless, we agree that this may influence generalizability of findings, and we added to the limitations section in lines 353-356: “We also excluded participants with 25.0-29.9kg/m2 BMI to ensure a difference in body size between groups as there can be considerable variation in body composition between individuals of similar BMI3 however this may influence the generalizability of our findings”.

  1. R-Table2, lines 166–167; R-text, lines 188–197: For KAA measurements with ICC values below 0.5, was any independent verification (e.g., video-based assessment) performed to confirm accuracy?  

Response: We agree that there is a need for independent verification and we apologize as this was not incorporated into our methods section. All 3D motion capture data were manually tracked and no errors in tracking or event identification were confirmed. The Kinect with ACL Gold software provides a screenshot of the frames identified as IC and PKF after each trial, which were visually examined at the time of data collection. Any trials showing misidentified events were discarded and the trials repeated. Further, participants with significant variation between 3D motion capture and Kinect were re-examined by an independent researcher. To clarify this process, we have added to lines 134-137: “3D motion capture data and still frames from the Kinect of IC and PKF were visually examined for missing marker data or event identification error. No marker tracking errors were identified, and Kinect trials with error in event identification or pose estimation were discarded.”

  1. M2.2, lines 83–85: Given that the analysis involved a 2×2×2 ANOVA, how was the sample size calculation adjusted to ensure adequate power for detecting interaction effects?  

Response: Thank you for this comment. We included the ANOVA as an additional analysis to interpret our research question because the ICC explains the extent to which data are similar, but does not clarify if they are different. Large effect sizes were expected based on previous studies comparing landing biomechanics between males and females, and between those with high and low BMI. As such, we estimated that 9 participants per group (36 total) assuming 80% power, 4 groups and 2 measurements per participant (See below using G.Power). We have added a comment into the methods section on this point in lines 92-95: “Additionally, 9 participants per group were required to achieve 80% power based on large effect size differences between males and females and between those with high and low BMI in a mixed factor ANOVA design (f=0.3, β=0.2, α=0.05; G.Power 3.1.9.7 - see attached word document for screenshot of output).”

  1. R-Table2, lines 166–167: How do the authors interpret ICC values with such wide confidence intervals, some spanning from negative values to high positive values?  

Response: We agree that the ICCs with wide confidence intervals require additional consideration. The wide confidence intervals suggest uncertainty in the true agreement between methods. which readers should be aware of. We have added a paragraph to the discussion at lines 292-302 to discuss the implications of wide confidence intervals, including negative values. The added paragraph discusses the reasons for wide confidence intervals and recommends awareness of the uncertainty of the true agreement between methods when interpreting Kinect-derived measurements in practice or in other literature. Further, at lines 299-302 we have identified the groups and timepoints with the narrowest confidence intervals to support the finding of moderate-to-good agreement between methods. “In our study, the groups with the narrowest confidence intervals were low BMI females at IC and low BMI males at PKF when measuring KASR, suggesting the true agreement between methods may be in the moderate-to-good range for these groups and timepoints”. We have also added “This supports the use of the Kinect as a measurement tool for measuring during landing although wide confidence intervals suggest some uncertainty on agreement between methods that practitioners should be aware of.” to lines 381-383 in the conclusion.

  1. C, lines 343–354: Can the authors clarify under which specific conditions the Kinect tends to underestimate injury risk, rather than presenting this as a universal outcome?  

Response: Thank you for identifying the lack of specificity in the conclusion. To improve the conclusion, we have edited lines 379-380 to state “The Kinect had the best agreement with 3D motion capture with ICCs in the good range for all females at IC, all males at PKF and high BMI females at PKF.” We also expanded the conclusion to specify when the Kinect underestimated KAA and how the underestimation may influence injury screening in lines 387-390 “The Kinect underestimated the KAA compared to 3D motion capture for females and males with high BMI at IC and PKF. The underestimation of KAA in the high BMI groups caused the Kinect to not identify the different landing patterns between high and low BMI groups that was identified with motion capture.”

  1. M2.2, lines 86–90; D, lines 308–312: How applicable are these findings to elite athletes or high ACL-risk populations, given that the sample consisted of recreationally active adults?

Response: You are correct in that our study included recreationally active adults rather than focusing on elite athletes. Recreationally active adults are also at risk of ACL injury when participating in activities. However, there is less homogeneity within the group than in elite athletes who may exhibit specific movement patterns based on their sport. We included a recommendation for further evaluation of the Kinect with activity level stratification that may identify differences between sedentary individuals, recreationally active and elite athletes in lines 370-375. Other than athletic participation, we considered high-risk demographic groups when designing this study, such as sex and body size. To clarify the relationship between sex, BMI and ACL injury risk, we have added to lines 56-58 to state “Analyzing landing biomechanics is important for individuals at greater risk for ACL injury such as individuals with high BMI or females while participating in physical activity.” Lines 59-72 further describe the relationship between sex, body size and ACL injury risk and discusses the implications for motion analysis.

Round 2

Reviewer 1 Report

Comments and Suggestions for Authors

The authors have revised the manuscript appropriately in accordance with the reviewers’ comments.

Reviewer 2 Report

Comments and Suggestions for Authors

The authors have adequately addressed all of my previous comments.
The revisions improved the clarity and quality of the manuscript.
I have no further concerns and recommend acceptance in the present form.